# Analysis of Infiltrating Water Characteristics of Permeable Pavements in a Parking Lot at Full Scale

**Jaerock Park [1], Jaehyun Park [1], Jonghyun Cheon [1], Jaehyuk Lee [2] and Hyunsuk Shin [1,\*]**

[1]  Department of Civil and Environmental Engineering, Pusan National University, Busan 46241, Korea; closej524@pusan.ac.kr (J.P.); kodam989@naver.com (J.P.); cjhcin8376@naver.com (J.C.)
[2]  Green Land and Water Management Institute, Pusan National University, Busan 46241, Korea; rasmania@hanmail.net
\*  Correspondence: hsshin@pusan.ac.kr; Tel.: +82-51-510-2348

**Abstract:** Impermeable materials are used for parking lots at apartment complexes and large stores which are concentrated in urban areas. These materials increase the amount of surface runoff by blocking infiltration, resulting in flood damage, dry stream phenomena in rivers in urban watersheds, and the depletion of ground water. In this study, a parking lot plot was constructed to quantitatively evaluate the efficiency of pavements using various materials (impermeable concrete, permeable concrete, and permeable block pavement). Four scenarios of rainfall intensity were simulated using a rainfall simulator within each plot (36 mm h$^{-1}$, 48 mm h$^{-1}$, 60 mm h$^{-1}$, 72 mm h$^{-1}$). The flow was observed by monitoring the system with a bucket flow meter. The efficiency and flow characteristics of the permeable concrete and block pavement were analyzed. The results were used to calculate the ratio of the surface flow to the infiltrating flow between impermeable and permeable pavements. The permeable concrete had a ratio of 1:0.9, and the permeable block pavement had a ratio of 1:0.58.

**Keywords:** low impact development; permeable pavement; infiltrated water quantity; infiltrate coefficient

## 1. Introduction

Recently, the frequency and intensity of rainfall in summer have been increasing in Korea due to climate change. As a result, the scale of flood-related damage has been increasing and effective disaster prevention measures are needed [1]. The typhoon "Rusa" generated nearly 900 mm of precipitation for two days in 2002 and caused massive losses, taking 321 victims and damaging 5.1479 trillion KRW (Korea Won) of property. In the following year, extensive damage caused by the typhoon "Maemi" (130 casualties and property damage estimated at 4.2225 trillion KRW) led to increased governmental awareness regarding flood prevention measures [2]. In addition to issues of time and cost, reinforcing standards for the sewerage system or adding retention facilities would not have been appropriate solutions to reduce the flood damage in the city. Two major Korean cities, Busan and Ulsan, still suffered the effects of inundation due to typhoon "Chava" (2015) and torrential rainfall in 2017 despite having capable drainage systems and a huge underground storage system in problem areas.

The sealing of soil can lead to a decrease in water permeability, a loss of biodiversity, and a reduction in the capacity of the soil to act as a carbon sink. Although it is evident that soil sealing has a strong impact on the soil resource, little direct evidence was found in the literature that can help quantify its influence [3]. Low Impact Development (LID) techniques have attracted attention in South Korea in recent years and are advantageous in various urban situations [4]. Permeable pavements, which were first implemented in the late 1990s, are one of the fundamental technologies of LID. This technology was developed to rehabilitate the hydrological cycle of urban watersheds to their

natural state, with the additional function of preventing flood inundation by reducing runoff and storing storm water from the impervious area. Many studies have examined the efficiency of permeable pavements and ways to improve their structural durability, which is lower than that of conventional paving materials, as well as clogging phenomena. The first step of this research concerns pavement technology (the subgrade materials and depth of layers) that is suitable for urban watersheds, which are mainly constructed with impermeable materials. The research then explores the effectiveness of permeable pavements using modelling programs or observations after the application of methods in a parking lot area. According to Benjamin [5], the superiority of permeable pavements in terms of durability, infiltration, and water quality was demonstrated in relation to the lifespan of constructions by observing the technical behavior of permeable and conventional asphalt pavements for six years. Daniel Jato-Espino simulated the optimal location of permeable pavements to reduce flood damage during storm events through the linkage of storm water modelling and GIS [6]. Lee analyzed the effects of reducing runoff through a permeable pavement facility. The study also demonstrated that permeable pavements provide an effective means of preventing flood disasters in lowland and downstream watersheds by observing the decreased peak flow and runoff volume after installing them in an urban watershed located in the lowlands [7]. Lim used a simplified model with XP–SWMM (Storm Water Management Model) and evaluated the effect of reducing runoff when replacing a conventional asphalt pavement with a porous pavement in the city. Andersen conducted an experiment concerning the relationship between permeable pavement drainages and their evaporation capacities. The experiment aimed to measure the specific movement of water components in structures over time. The results demonstrated that evaporation, drainage, and water retention in structures were strongly influenced by the particle size distribution of the bedding material and by surface storage effects in the surface blocks [8]; however, the experiment only focused on the pervious blocks and their evaporation on a laboratory scale [9]. Cipolla conducted a field investigation that compared the infiltration rates of eight permeable parking lots by the single ring test method. The infiltration rates were mostly affected by the position of the test ring, the subgrade material, and the surface type rather than by the antecedent dry time, pavement age, and pavement compaction with regard to the diminution of the infiltration rates; however, the result did not consider the differences of subsurface layers and slopes [10].

Previous studies have focused on observing the performance of permeable pavements and the changes in the runoff characteristics of particular sites with different areas and slopes by either installing permeable pavements or by using a modelling program. The experiments, under all the applied conditions, could not quantize or apply a suitable design or field for permeable pavement. This study analyzes the behavior of the infiltrated water and analyzes a part of the water cycle by using different permeable packages in small watersheds of the same slope and width.

(1)　We analyzed the effects on rainwater management and surface flow reduction when permeable pavements were installed in parking lots, which are a significant portion of the impermeable area in urban watersheds.

(2)　All of these facilities were full-scaled, same-sized lots consisting of sand, aggregate and a bedding layer for pavements with an infiltrated water monitoring system. The same rainfall scenarios were simulated for impermeable and permeable pavement plots (concrete and block), located in the parking lot LID observation facility of the Korea Green Infra (GI) and Low Impact Development (LID)Centre, Pusan National University, Busan, South Korea. See Figure 1 below for location and coordinates.

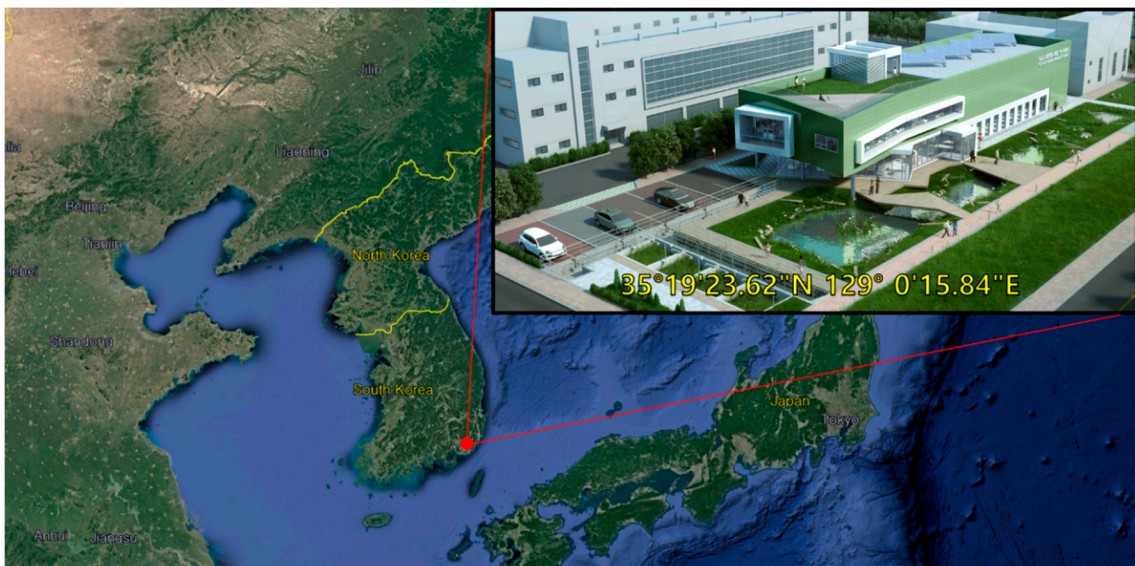

**Figure 1.** Parking lot LID observation facility of the Korea GI and LID Centre, Pusan National University, Busan, South Korea (35°19′23.62″ N 129°0′15.84″ E).

## 2. Materials and Methods

### 2.1. Parking Lot LID Facility

Impermeable materials prevent rainwater from infiltrating the urban watershed, thus causing a hydrological cycle imbalance. Permeable pavements were developed to restore the hydrological cycle in the urban watershed to its natural state before urbanization. They have the function of preventing disasters such as urban floods by reducing runoff in the city and storing rainwater.

The parking lot LID observation facility was designed with a concrete box (area: 10.85 m × 2.3 m, height: 0.9 m) and a slope of 1% to help with smooth drainage (Figure 2). The cross-section of the impermeable concrete pavement (ICP) consisted of a Portland-cement surface layer and a crushed-aggregate sublayer (D13, 25 mm). The permeable concrete pavement (PCP) was composed of a Bio-pave surface layer that was mixed with vegetable polyurethane resin in the aggregate with a diameter of less than 10 mm. There was also a filter layer composed of fine aggregate to improve infiltration and water quality, as well as a blanket layer to protect against frost damage by forming a capillary barrier. The permeable block pavement (PBP) was composed of blocks, filling materials, and a stable layer. It consisted of two types of crushed aggregate at the lower surface layer and a sand filter to improve the water quality. The details of the cross-section of the three plots are shown in Figure 3. The internal structure of the section of the parking lot was designed with reference to the standards of American Association of State Highway and Transportation Officials (AASHTO) [11,12].

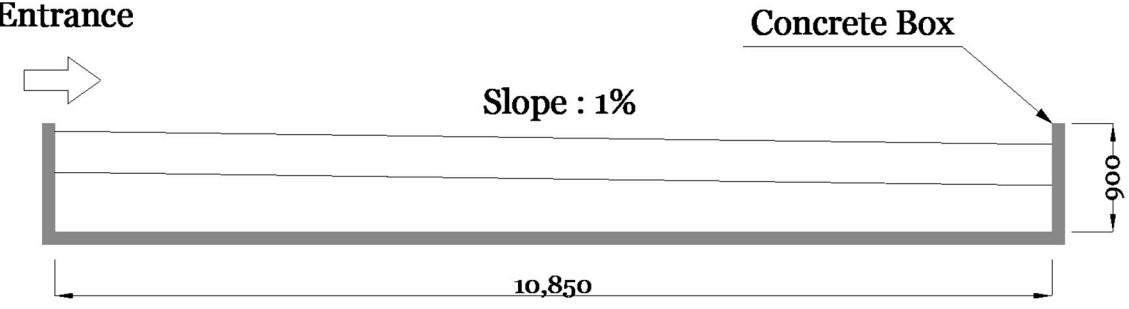

**Figure 2.** Longitudinal section of Parking Lot LID Facility. All of the parking lot has a 1% slope (Length unit: mm).

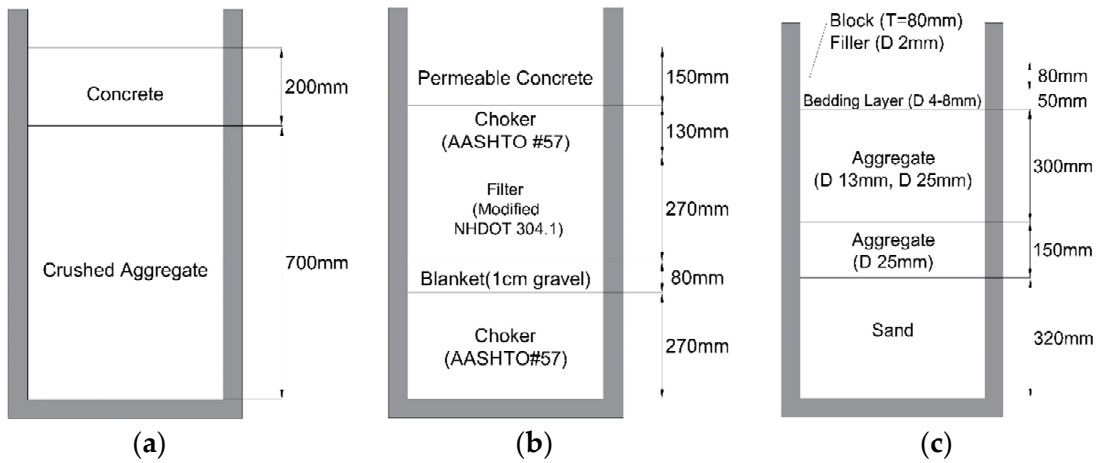

**Figure 3.** Longitudinal section of each pavement: (**a**) Impermeable concrete, (**b**) Permeable concrete (infiltration rate: 9.45~12.12 mm/s) with sand filter, (**c**) Permeable block (infiltration rate: 1.53~3.64 mm/s).

## 2.2. Scenario

The ICP was used as a control, and the experimental groups were PCP and PBP. We based the area average rainfall intensity for each scenario used in the experiment on the 60 min probable rainfall intensity of Busan, Korea. The reason for choosing a duration of 60 min was to easily grasp the behavior of water circulation in the small watershed through sufficient rainfall and duration. In addition, the effect of permeable pavements on small flashes in urban watersheds can be seen through the small return period selection. The return period for each scenario is shown in Table 1 below.

**Table 1.** 4 Scenarios for Rainfall Simulations.

|  | Case 1 | Case 2 | Case 3 | Case 4 |
|---|---|---|---|---|
| Simulating Intensity (mm h$^{-1}$) | 36 | 48 | 60 | 72 |
| Return Period (year) | 2 | 3 | 5 | 10 |
| Busan (mm h$^{-1}$) | 43.02 | 51.26 | 60.44 | 72.06 |

## 2.3. Movable Rainfall Simulator

A movable rainfall simulator (Figure 4) was used to generate the required rainfall intensity with no restrictions on location. The simulator consisted of several cubic frames, two pumps, a 1000 L water tank, nozzles, flow meters, and bypass and wind shields to minimize wind effects.

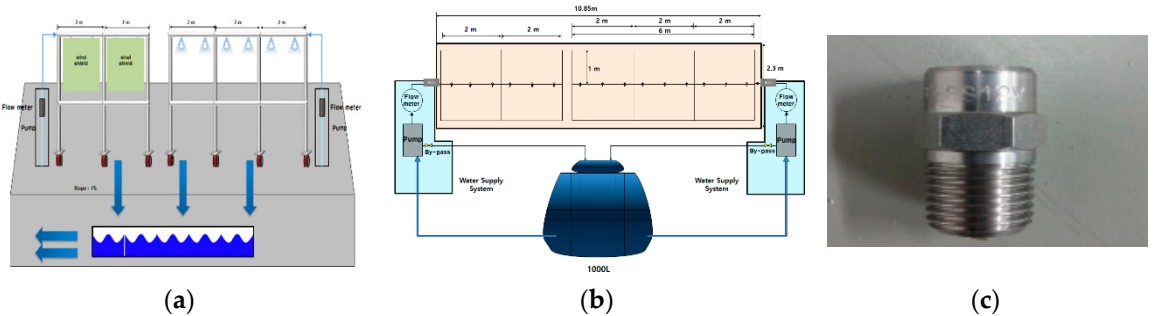

**Figure 4.** Rain Simulator and Nozzle. (**a**) Diagram of the whole design, (**b**) Rain simulator system with test bed, (**c**) KJ 1/4 FF-SS 12W nozzle. Orifice diameter: 3.3 mm, Angle of injection pattern: 120°, Water particle size: 0.51–3 mm, Injection flow rate: 9.8 L/min.

The coverage of each cubic frame was 4 m$^2$ (2 m × 2 m) in area with a height of 2.6 m. They were assembled in a series to cover the experimental site. To verify the rainfall intensity of the simulator, a pre-experiment was conducted to develop the relationship between inflow and the relevant rainfall intensity with nine rain gauges installed under the simulator.

*2.4. Monitoring*

In the ICP, i.e., one of the simulated scenarios, measurements were obtained until 20 min after the end of the rainfall simulation to measure the surface runoff. After that, we simulated the same scenarios for the two permeable pavements (PCP, PBP) and measured the runoff for 5 h after the end of the rainfall simulation to establish the infiltration. The Impermeable Concrete-Surface Flow-Monitoring Box, the Permeable Concrete-Infiltrated Flow-Monitoring Box, and the Permeable Block-Infiltrated Flow-Monitoring Box all use a tipping bucket to take measurements (shown in Figure 5, capacity: 25 L/min). The flow rate measured by the monitoring box was recorded and stored every minute at the monitoring system of Korea GI and LID Center.

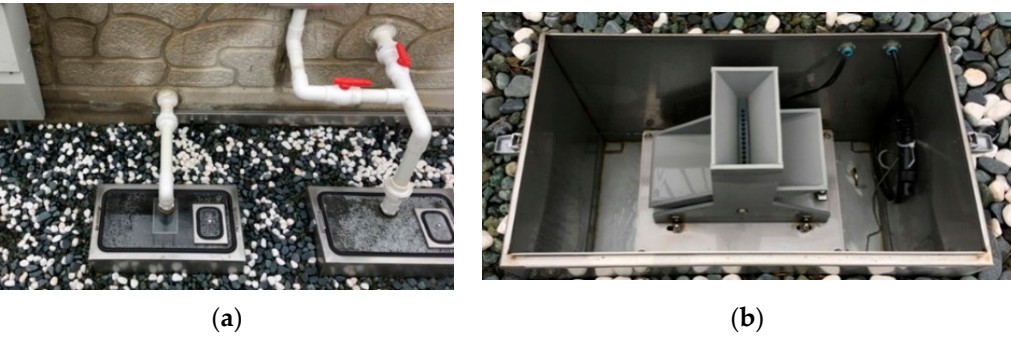

(**a**) (**b**)

**Figure 5.** Pavement systems with two monitoring boxes. (**a**) Infiltration water monitoring on the left, and surface run off on the right. (**b**) Both monitoring boxes used a 25 L/min tipping bucket.

**3. Results**

*3.1. Flow Characteristics*

In all scenarios, the impermeable concrete pavement displayed surface flow immediately after rainfall simulation began. For each scenario, as shown in Figure 6, a parallel section appeared after 12 min, 9 min, 10 min, and 8 min. Peak flows reached 11 L/min, 14 L/min, 17 L/min, and 19 L/min, and were maintained during the rainfall simulation. The surface flow began to decrease rapidly from the end of the rainfall simulation, and the runoff ended in all scenarios within 20 min after the end of the test. The total outflows for each scenario were 647 L, 803 L, 968.5 L, and 1187.5 L. The loss rate was calculated, using Equation (1), as 28.11%, 33.08%, 35.43%, and 34.03%.

$$loss\ rate = \frac{\Sigma Q_{in} - \Sigma Q_{out}}{\Sigma Q_{in}} \times 100\ (\%) \tag{1}$$

Surface runoff did not occur in all cases for the permeable concrete pavement (PCP), but infiltration flow began around 10 min after the start of the test. For each scenario, a parallel section appeared after 40 min, 40 min, 44 min, and 41 min. Peak flows reached 10 L/min, 12.5 L/min, 16 L/min, and 16 L/min, and the flow was maintained for 5 min after the end of the rainfall simulation. After that, the infiltrating flow began to decrease slowly, and the runoff ended in all the scenarios within 300 min after the end of the test. The total outflows in each scenario were 625 L, 778.5 L, 954.5 L, and 994.5 L. In the case of the permeable block pavement (PBP), surface runoff did not occur as in all cases of the PCP, but infiltrating flow began around 20 min after the start of the test. Stable flow occurred at 48 min, 50 min, 56 min, and 56 min. Peak flows reached 7 L/min, 7.5 L/min, 10 L/min, and 11 L/min. The peak flow was maintained for 30 min after the end of rainfall simulation. After that, the infiltrating flow began to

decrease but did so more slowly than that of the PCP; the runoff ended in all scenarios within 300 min after the end of the test. The total outflows for each scenario were 484.5 L, 750.0 L, 936.0 L, and 1087.5 L. Table 2 summarizes the infiltrated water flow data.

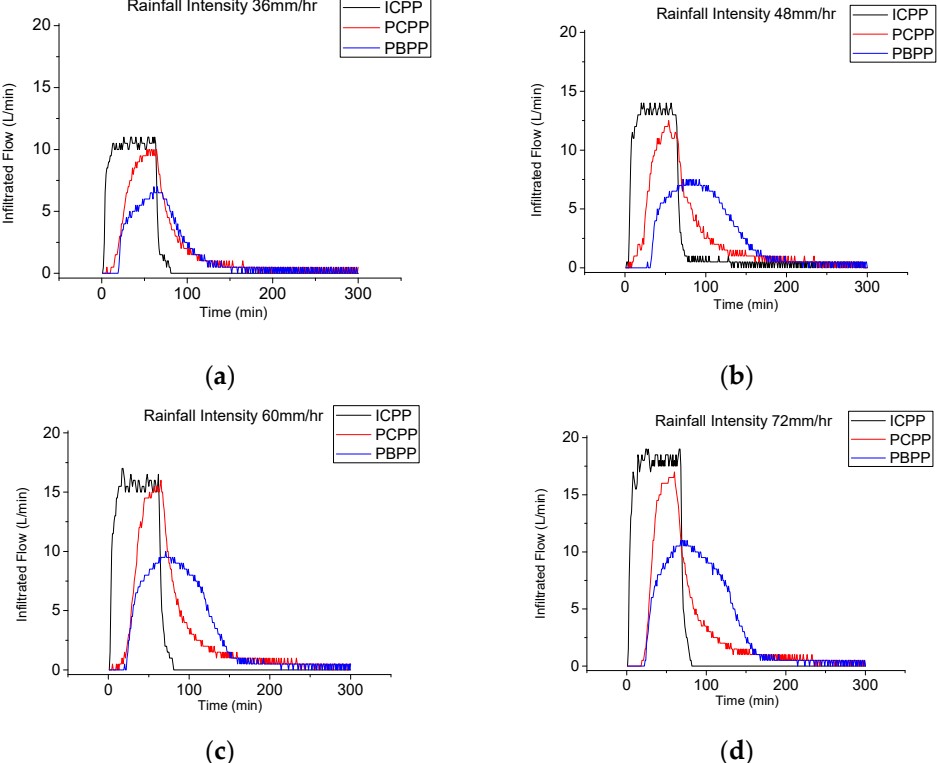

**Figure 6.** Comparing flow at three types of pavement in different scenarios: (**a**) 36 mm h$^{-1}$, (**b**) 48 mm h$^{-1}$, (**c**) 60 mm h$^{-1}$ and (**d**) 72 mm h$^{-1}$ (ICPP has the same unit and values as the Y-axis except for Surface runoff).

**Table 2.** Ratios of peak runoff to infiltration flow.

| Case | Travel Time(min) | | | Peak Flow(L/min) | | | Total Flow(L) | | |
|---|---|---|---|---|---|---|---|---|---|
| | ICP | PCP | PBP | ICP | PCP | PBP | ICP | PCP | PBP |
| 1 | 12 | 40 | 48 | 11 | 10 | 7 | 647 | 623.5 | 484.5 |
| 2 | 9 | 40 | 50 | 14 | 12.5 | 7.5 | 803 | 778.5 | 750 |
| 3 | 10 | 44 | 56 | 17 | 16 | 10 | 968.5 | 954.5 | 936 |
| 4 | 8 | 41 | 56 | 19 | 16 | 11 | 1187.5 | 994.5 | 1087.5 |

\* ICP: Impermeable Concrete Pavement, PCP: Permeable Concrete Pavement, PBP: Permeable Block Pavement.

### 3.2. Runoff Reduction Ratio

The average travel time until the peak flow for the PCP was 4.33 times higher than that of the ICP. The peak flow decreased by 9.1%, 10.7%, 5.9%, and 15.8%, and the total outflow decreased by 3.6%, 3.1%, 1.4%, and 16.3%, respectively. In the case of the PBP, the average travel time until the peak flow was 5.54 times higher than that of the ICP. The peak flow decreased by 36.4%, 46.4%, 41.2%, and 42.1%, and the total outflow decreased by 25.11%, 6.6%, 3.4%, and 8.4%, respectively.

We calculated the ratios between the peak flow caused by the surface flow on the ICP and the peak flow caused by the infiltration of the PCP and PBP. In the PCP, infiltrating flow occurred at ratios of 1:0.91 (Scenario 1), 1:0.89 (Scenario 2), 1:0.94 (Scenario 3), and 1:0.84 (Scenario 4). For the PBP, infiltrating flow occurred at ratios of 1:0.64 (Scenario 1), 1:0.56 (Scenario 2), 1:0.59 (Scenario 3), and 1:0.58 (Scenario 4). Table 3 summarizes the ratios of infiltration flow to impermeable pavement.

**Table 3.** Ratios of infiltration flow to impermeable pavement.

| Scenario | ICP | PCP | PBP |
|:---:|:---:|:---:|:---:|
| 1 | 1 | 0.91 | 0.64 |
| 2 | 1 | 0.89 | 0.56 |
| 3 | 1 | 0.94 | 0.59 |
| 4 | 1 | 0.84 | 0.58 |

## 4. Discussion

In general, Darcy's law can be used to predict the behavior of permeates in permeable materials such as soil. In the case of permeable pavements, various factors that can affect permeability should be considered [13]. In the case of water permeability, the compaction and particle size of the soil are relevant factors. As the compaction becomes larger, the permeability decreases, which may affect the permeability of the permeable pavement. This can increase the surface runoff, even on the soil surface, and put a heavy load on the sewerage systems in urban watersheds [14].

In the ICP, the runoff occurred at the same time as the start of the rainfall simulation, the peak runoff increased in proportion to the runoff, and the runoff decreased rapidly after the end of the rainfall simulation. We measured the outflow using the monitoring system and calculated the loss ratio by using the difference between the total inflow value and the total outflow value. As a result, around a 28~35% loss occurred during the experiment. In the case of PCP, all scenarios had no surface runoff and the infiltrated flow started around 10 min after the rainfall simulation. The total flow decreased by 1.4~16% more than in the case of the ICP. The travel time to the peak flow increased by about 4.33 times, and the peak flow decreased by 6~16%. In the case of the PBP, no runoff occurred, and infiltrated water flow started after about 20 min of the rainfall simulation. The total flow decreased by 3.4~−25.4%, and the travel time to peak flow increased by about 5.54 times compared to the ICP results. The peak flow decreased by 36.4%~46.4%. In this study, the fact that the parking lot-type LID observation facility was installed in a concrete box made it difficult to consider the decrease of the total flow and the storage capacity of the permeable pavement. If a permeable pavement is installed in a parking lot, it will delay the start of the flow and the travel time by inhibiting the surface flow, and can reduce the peak flow and total flow. Therefore, it can act as a countermeasure against urban flooding resulting from to heavy rainfall. In addition, through the use of a permeable pavement, rainwater can flow into the groundwater to reduce the wash off of nonpoint source pollutants caused by rainfall, thereby reducing river pollution [15,16].

## 5. Conclusions

Existing studies have concentrated on the behavior of water on the surface of permeable pavements, but few studies have focused on the behavior of infiltration water and the water quantity after infiltration. We simulated four rainfall scenarios using ICP, PCP, and PBP. Using the results of the simulation, we analyzed the characteristics and the reduction of the runoff on the two types of permeable pavements. Based on the experimental results, we calculated the ratio of the surface flow occurring on the ICP to the infiltrating flow in the PCP and PBP when simulating the same rainfall levels (ICP: 1:0.9, PBP: 1:0.58). The calculated ratios for impermeable concrete, permeable concrete, and permeable block concrete could be estimated by further experiments and research. In terms of water management, the same type of parking lot used in this experiment could be applied to a new city design or an existing city's parking lots. Thus, efficient design guidelines could be determined, such as the areas and locations of permeable pavement parking lots, which would be done by calculating the infiltrating flow volume based on the inflow volume using the estimated ratios. Because of the capacity limit of the tipping bucket (25 L/min), the sample data were limited because only four scenarios could be tested based on the 60 min probability of rainfall intensity in Busan. In addition, the cross section of the parking lot used in the experiment could be varied depending on the actual installation site, which may affect the

behavior of the infiltrate. In the case of the permeable pavement used in the experiment, there was no use period, which means that there was no consideration of clogging due to suspended solids and contaminants. The change in water flow due to clogging also affects the neighboring area [17], and for this, further research on the change in infiltration water behavior due to clogging is necessary. In addition, soil compaction and particle size were not considered when installing the soil in the support stabilization layer; therefore, studies on the change in the hydraulic conductivity of permeable pavements will be required, depending on the nature of the soil [18,19].

However, since the errors between the results are not particularly significant, this study could be used as a preliminary reference to estimate the ratio of the surface flow to the infiltrating flow through additional experiments and studies. Furthermore, depending on the treatment of infiltration water, it is expected that it may be used for various rainfall management techniques using permeable pavements, such as the prediction of water quality when considering the inflow of groundwater, and the calculation of capacity when installing storage tanks.

**Author Contributions:** Conceptualization, J.P. (Jaerock Park) and J.P. (Jaehyun Park); methodology, formal analysis and validation, Jaerock P., J.P. (Jaehyun Park) and J.C.; writing—original draft preparation, J.P. (Jaerock Park), and J.P. (Jaehyun Park); writing—review and editing, J.P. (Jaerock Park) and J.L.; supervision, J.L. and H.S.; funding acquisition, H.S. All authors have read and agreed to the published version of the manuscript.

**Funding:** This research was supported by a grant (2016000200003) from Public Welfare Technology Development Program funded by Ministry of Environment of Korean government.

**Conflicts of Interest:** The authors declare no conflict of interest.

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
