# Peer review of "Analysis of Infiltrating Water Characteristics of Permeable Pavements in a Parking Lot at Full Scale"

_water, doi:10.3390/w12082081_

Round 1
Reviewer 1 Report
Well-done paper concerning the use of Permeable Pavements in a Parking Lot at Full Scale.
Introduction part is quite well done, however, deeper literature overview should be done. All paper based on 10 references. In this topic it is not enough.There should be mentioned more papers and studies which were namely dealing with water quality in terms of storm water infiltration on parking.
The manuscript concerns quantitative research, however, in the subject matter it is necessary to mention qualitative issues Even in the issues of the necessity of conducting further research using this type of pre-permeable surface
Additional suggestions:
Materials and Method
Figure 1. - Add units and check dimensions on figure 1.
I'm not sure but based on page 3 it should be 10850mm?
In the line 83 i suggest to describe and add references “on some assessment index methods released previously“ - point which one
Scenario
This section is good described but I wonder why you choose only rainfall duration 60 minutes ? For small urban floods, shorter duration rainfall is often critical.
The conclusions are described correctly, however, there is no reference to the quality of rainwater.
If such studies have not yet been conducted (having such facilities) it would be very good to describe as a further direction of research.
Reviewer 2 Report
Water845925 Analysis of infiltrating water characteristics of permeable pavements in a parking lot at full scale
The work presents extremely interesting data. The comparative discussion is completely missing. The conclusions require an extrapolation, in order to be understood even by non-specialists. However, the Authors should make their discussion and conclusion sections more incisive, discussing, comparatively, the practical feasibility of the proposed approach. More literature on the same topic must be addressed.
This manuscript adheres to the journal’s standards. The research meets the applicable standards for the research integrity. The article does not adhere to appropriate reporting guidelines and community standards for data availability: the complete raw database is not fully made completely available in a public repository, such as Zenodo, for instance.
The research output, in terms of novelty, scores very good uniqueness in terms of data. The level of clarity is by the threshold of acceptability. It does adopt up to date methodologies in respect to the object of research. The paper does not discuss convincingly the limitations of the approach and potential biases due to the assumptions made.
Potentially, its potential impact upon the international scientific community of reference, as it is, is discrete. The study presents the results of primary scientific research. Experiments, statistics, and other analyses are performed to a sound technical standard. Conclusions presented are supported by the data however their level of novelty must be extrapolated and enhanced.
The article is presented in an intelligible manner. This work is interesting and deserves to be published after accurate revision.
Title: OK
Abstract: OK
Acronyms: Many (too many) acronyms are used. Sometimes, without spell them out, the first time they are introduced into the text. Inserting a list of 'Abbreviations' would facilitate the reader.
Keywords: REVISE. The keywords, together with title and abstract function in a system comparable to a chain reaction. Once the keywords have assisted the Reader find the suitable paper and its title has fruitfully drawn in the attention, it is up to the abstract to further activate the interest and keep their curiosity. So, these three elements must work together and not replicate each other.
Introduction: REVISE. Please, focus the paper main aim.
Method: REVISE. Some materials or methods are inadequately discussed. Please, refer to ISO methods
Results: OK
Discussion: REVISE
Conclusion: REVISE
References: REVISE
In particular (page.row):
1.18 mm h-1
2.54 please, spell out acronyms
3.88 geographical coordinates are mandatory
7.177 soil sealing is the permanent covering of the land surface by buildings, infrastructures or any impermeable artificial material. It has been identified as a major threat in the Soil Thematic Strategy of the European Commission, both in terms of permanent loss of soil as a resource and for its important impacts on soil functionality. The importance of soil sealing in urban areas is perceived as a driver of flood risks in many contexts (doi>10.1016/j.landurbplan.2008.10.011). Please, extrapolate your intriguing results comparatively.
Recommendation: Major revision is required
Reviewer 3 Report
1) The authors use the term watershed to describe the experimental plots. It would be more appropriate to describe he study as a plot experiment.
2) the bulk of the editing for English grammar is in the introduction lines 26 to 64.
Reviewer 4 Report
I would urge the authors to explictly state the innovativeness of this research.
Round 2
Reviewer 2 Report
- The work presents extremely interesting data. The comparative discussion is completely missing. The conclusions require an extrapolation, in order to be understood even by non-specialists. However, the Authors should make their discussion and conclusion sections more incisive, discussing, comparatively, the practical feasibility of the proposed approach. More literature on the same topic must be addressed.
- The limitations and validity of the experiment were presented. In addition, there have been many previous papers on the infiltration rate of permeable pavement, but there have been no studies on the behavior of infiltrated water and comparative analysis was impossible.
- This is simply not true. A rapid Scopus search (14.07.2020) scores 38 documents (TITLE (soil AND sealing) AND TITLE-ABS-KEY (infiltration)). Please, discuss them comparatively. This will make the manuscript more interesting.
- This manuscript adheres to the journal’s standards. The research meets the applicable standards for the research integrity. The article does not adhere to appropriate reporting guidelines and community standards for data availability: the complete raw database is not fully made completely available in a public repository, such as Zenodo, for instance.
- Uploaded the dataset to Zenodo.
URL: https://doi.org/10.5281/zenodo.3938060 - Happy for this upload, but I cannot open it. Please, upload a rougher but also more accessible file
- 88 geographical coordinates are mandatory
- The phrase mentioned has been deleted.
- geographical coordinates are mandatory at row 79
- 177 soil sealing is the permanent covering of the land surface by buildings, infrastructures or any impermeable artificial material. It has been identified as a major threat in the Soil Thematic Strategy of the European Commission, both in terms of permanent loss of soil as a resource and for its important impacts on soil functionality. The importance of soil sealing in urban areas is perceived as a driver of flood risks in many contexts (doi>10.1016/j.landurbplan.2008.10.011). Please, extrapolate your intriguing results comparatively.
- It was added to the conclusion by referring to the presented literature. Please see the below
- Sorry, but I do not see any of the declared additions, not text neither quotations (the version I’ve downloaded contains 10 quotations)
Author Response
Thanks for the detailed review, Please see the attachment.

Round 3
Reviewer 2 Report
Row 87: geographical coordinates are mandatory
